# Dynamic Trends in Sociodemographic Disparities and COVID-19 Morbidity and Mortality—A Nationwide Study during Two Years of a Pandemic

**DOI:** 10.3390/healthcare11070933

**Published:** 2023-03-23

**Authors:** Arielle Kaim, Mor Saban

**Affiliations:** 1Health Technology Assessment and Policy Unit, The Gertner Institute for Epidemiology & Health Policy Research, Sheba Medical Center, Tel-HaShomer, Ramat-Gan 5266202, Israel; 2Department of Emergency & Disaster Management, School of Public Health, Sackler Faculty of Medicine, Tel-Aviv University, Tel-Aviv-Yafo 6139001, Israel; 3Nursing Department, School of Health Professions, Sackler Faculty of Medicine, Tel-Aviv University, Tel-Aviv-Yafo 6139001, Israel

**Keywords:** health trends, socioeconomic gradient, pandemic, disease

## Abstract

Social epidemiological research has documented that health outcomes, such as the risk of becoming diseased or dying, are closely tied to socioeconomic status. The aim of the current study was to investigate the impact of socioeconomic status on morbidity, hospitalization, and mortality outcomes throughout five waves of the pandemic amongst the Israeli population. A retrospective archive study was conducted in Israel from March 2020 to February 2022 in which data were obtained from the Israeli Ministry of Health’s (MOH) open COVID-19 database. Our findings, though requiring careful and cautious interpretation, indicate that the socioeconomic gradient patterns established in previous COVID-19 literature are not applicable to Israel throughout the five waves of the pandemic. The conclusions of this study indicate a much more dynamic and complex picture, where there is no single group that dominates the realm of improved outcomes or bears the burden of disease with respect to morbidity, hospitalization, and mortality. We show that health trends cannot necessarily be generalized to all countries and are very much dynamic and contingent on the socio-geographical context and must be thoroughly examined throughout distinct communities with consideration of the specific characteristics of the disease. Furthermore, the implications of this study include the importance of identifying the dynamic interplay and interactions of sociodemographic characteristics and health behavior in order to enhance efforts toward achieving improved health outcomes by policymakers and researchers.

## 1. Introduction

Social epidemiological research has documented that health outcomes, such as the risk of becoming diseased or dying, are closely tied to socioeconomic status [1,2]. Despite the variation and change geographically in social epidemiological patterns over time, for a great number of diseases and causes of death, the literature has widely stated that becoming sick or experiencing premature death is a risk that increases with a lower socioeconomic status; however, the exact mechanisms underlying this relationship are complex and multifaceted [3,4]. Health inequality trends have been revealed even in countries in which modern welfare systems exist. In particular, chronic diseases and chronic infectious disease distributions have shed light on the differences in the frequency and severity of disease between socioeconomic groups [4,5]. In the context of viral respiratory diseases, analyses of the 1918, 1919, and 2009 influenza pandemics showed a greater risk of contracting the disease and dying among socioeconomically disadvantaged populations [6,7].

In the ongoing public health COVID-19 crisis, first identified in Wuhan, China, various risk factors impacting population outcomes have been exposed. Through the surveillance of clinical characteristics and hospital course outcomes of laboratory-confirmed cases of the virus, it was quickly identified that risk factors for more severe course of disease include both older age and the presence of a comorbid disease [8,9]. Findings from international literature globally have ascribed evidence of disparities based on socioeconomic status, whereby those who bear the burden of the pandemic are predominantly from the lower socioeconomic groups (Findings from North America [10,11]; Europe [12,13,14]; South America [15,16]; Africa [17,18,19]; Asia [19,20]; Australia [21]). In addition to socioeconomic status, other factors, such as profession, have been identified as risk factors for higher disease severity. For instance, healthcare professionals have been found to be at a greater risk of contracting and experiencing severe symptoms of infectious diseases, including COVID-19, compared to the general population [9].

Understanding the dynamics and effects of societal risk factors that make some groups particularly vulnerable is an essential part of ensuring more effective mitigation interventions during the ongoing and future pandemics.

Israel has a population of more than nine million people with diverse socioeconomic and demographic subpopulations [22]. The breakdown of the Israeli population composition is approximately 74% Jewish, 21% Arab, and 5% belonging to other ethnicities [22]. Among the Jewish population, approximately 12% belong to a distinct subpopulation which is religiously ultra-Orthodox. As compared to the general Jewish population, both the Arab and ultra-Orthodox Jewish populations are characterized by having a lower socioeconomic status (SES), higher fertility rates, and are younger [22,23,24]. Research has shown that municipalities with a lower socioeconomic status (SES) often experience reduced access to healthcare resources, which can result in poor health outcomes and lower uptake of preventive measures to safeguard public health [24]. This is often due to a lack of funding and infrastructure necessary to support healthcare delivery in these areas. However, it is worth noting that, in Israel, the National Health Insurance Law provides universal health insurance coverage to all citizens, regardless of their socioeconomic status [25]. This means that every Israeli resident has access to outpatient and inpatient healthcare services, including primary care, specialist consultations, hospitalization, and prescription medication, among others.

The universal healthcare system in Israel has contributed to significant improvements in health outcomes and has increased life expectancy in the country. The system is largely funded through a progressive tax system that ensures the wealthier population contributes more to healthcare than those with lower incomes, thereby addressing some of the health inequities associated with socioeconomic status. Additionally, the system emphasizes preventative care, which has led to lower rates of chronic diseases, better management of acute illnesses, and improved maternal and child health outcomes. The aim of the current study is to investigate the impact of socioeconomic status on morbidity, hospitalization, and mortality outcomes throughout five waves of the pandemic, specifically in the context of the Israeli population, in which there had been over ten 4.8 million confirmed cases and over 12,000 deaths as of 7 March 2023 [26].

## 2. Methods

### 2.1. Data Collection

A retrospective archive study was conducted in Israel from March 2020 to January 2022. Data were obtained from the Israeli Ministry of Health’s (MOH) open COVID-19 database, which includes information on 281 medium or large (1493 inhabitants or more) urban localities. COVID-19 information by socioeconomic (SE) status is not available at the individual level, and therefore, place of residence was used as an acceptable, widely used proxy [27]. The analysis included all 281 localities, which included 9,183,559 million residents, comprising 98.8% of the total population of 9.291 million.

The database contains national data on the number of COVID-19 diagnostic tests performed (excluding tests for recovered people), the number of confirmed cases (i.e., those who tested positive by real-time quantitative reverse-transcriptase polymerase-chain-reaction (qRT-PCR) assay. A person who tested positive was confirmed to be infected with COVID-19 regardless of the presence of any clinical symptoms (being symptomatic) and reoccurrence cases were removed from the dataset), the number of hospitalizations, and the number of deaths.

### 2.2. Data Analysis

In the first phase of the analysis, the data were split into five waves [(Wave 1: February–May 2020), (Wave 2: June–October 2020), (Wave 3: November 2020–March 2021), (Wave 4: April–October 2021), (Wave 5: December 2021–February 2022)] according to the definition of the Israeli MOH. Following this, we conducted a descriptive analysis and, for each wave, computed the number of qRT-PCR assay tests performed, the number of confirmed cases, the number of severe illness cases, and the number of deaths. Furthermore, we computed the ratio between deaths and confirmed cases (case fatality ratio) and the ratio between cases of severe illness and confirmed cases (severe case fatality ratio). In addition, we assessed the highest number of confirmed cases per day, the highest number of deaths per day, the highest number of severe cases per day, the highest number of tests conducted, the highest percentage of positive tests per day, and the highest number of active cases per day.

Following this, we linked each locality in the MOH database to its socioeconomic (SE) cluster. SE clusters are homogenous units ranked on a scale of 1 (lowest) to 10 (highest) determined by the Central Bureau of Statistics (CBS [28]) The area-level SES measure, available from 2012, is based on small statistical areas used in Israel’s census. The Central Bureau of Statistics uses information on demographics, education, employment, housing conditions, and household income to define the small statistical areas. The bureau uses a factor analysis to obtain a robust and valid measure reflecting the multidimensional nature of SES at the area level.

SE clusters are further grouped by the CBS into four categories derived from the Israel National Program for Quality Indicators [28] with 1 as the lowest SE ranking (clusters 1–3), followed by 2 (clusters 4 and 5), 3 (clusters 6 and 7), and 4 as the highest SE ranking (clusters 8–10) [29]. Categories 1, 2, 3, and 4 account for 28.5%, 18.3%, 27.2%, and 25.9%, respectively, of the Israeli population. These data were then age-adjusted according to three groups (0–39, 40–59, 60+) and divided by the five waves [(Wave 1: February–May 2020), (Wave 2: June-November 2020), (Wave 3: December 2020–April 2021), (Wave 4: May–November 2021), (Wave 5: December 2021–February 2022) [26]. Accordingly, the age-adjusted SES data were utilized for the calculation of confirmed cases, hospitalization cases, and death cases. The measures were calculated as the rate per 100,000 individuals in the population.

We then calculated the relative risk (clusters 1, 2, and 3 compared to cluster 4) of hospitalization and death and the confirmation rate for each wave. We analyzed the proportional differences in the numbers of confirmed, hospitalized, and death cases in each SES within and between waves using Z test for discrete data. All statistical analyses were performed using SPSS version 28 IBM SPSS 28.0 Statistics (IBM Corp. Released 2021. IBM SPSS Statistics for Windows, Version 28.0. IBM Corp., Armonk, NY, USA). *p*-values lower than 0.05 were considered to be statistically significant.

## 3. Results

Table 1 summarizes the morbidity and mortality findings for each wave.

The total number of confirmed cases was higher in the fifth wave as compared to the others. The ratio between the number of confirmed cases in the first wave compared with that in the fifth wave (February to May 2020) was almost 70 times higher. The positive rate of testing (22.21%) was also higher in the fifth wave when compared to previous waves.

However, the number of severe patients was quite similar to those in wave numbers two and four and lower than that in wave number three. The ratio between the number of confirmed cases and the number of severe illness cases was the lowest in the fifth wave (0.0003). A similar trend was found for the mortality rate, whereas in the fifth wave, the ratio between confirmed cases and death was 0.00009 (7.5 times lower than in the fourth wave), despite the number of confirmed cases per day being 26.9 times higher than in the fourth wave.

During the study period, 49,498,031 diagnostic tests were performed, and 3,480,823 confirmed cases were identified (overall 18.73% positive tests, reaching peaks of 10.89%, 15.52%, 10.19%, 8.42%, and 22.96% in the first, second, third, fourth, and fifth waves, respectively).

During the second and third pandemic waves, the highest rates of positive tests occurred among the lowest socioeconomic (SE) clusters (1–3), with intermediate rates in clusters 4–7 (4–5 -> cluster 3; 6–7 -> cluster 4), and the lowest rates in the highest (8–10) clusters (Figure 1). In October 2020 (the peak of the second pandemic wave), the age adjusted rates per 100,000 population of positive tests were 3415.5 and 1881.6 for SE clusters 1 and 2 and 1628.2 and 117.6 in clusters 3 and 4, respectively. In February 2021 (the peak of the third wave), the age adjusted rates per 100,000 were 7766.0 and 3965.0 in clusters 1 and 2 and 2319.0 in cluster 4. A similar picture was seen in the first and fourth waves but with smaller SE differences. During the fifth wave, the highest rates of positive tests occurred among the highest SE, level 4 (19,933.5 per 100,000 population).

Figure 2 depicts the percentages of hospitalization for the four SES categories for each wave. During the first wave, the proportion of new hospitalizations was the highest among SE cluster 4 (81.6 per 100,000). During the second and third waves, the highest proportion of new hospitalizations was observed among SE cluster 1 (142.6 and 256.8 per 100,000, respectively). The highest proportion of new hospitalizations in the fourth wave occurred among cluster 3 (131.8 per 100,000). During the fifth wave, the highest proportion of new hospitalizations occurred among SE clusters 3 and 4.

No such clear gradient was observed for COVID-19-associated mortality when analyzed by SE cluster (Figure 3). During the first wave, the highest percentage of mortality was observed in SE cluster 1 (3.3 per 100,000), while in both SE clusters 3 and 4, no deaths were recorded. During the second wave, the highest percentage of mortality was observed among SE clusters 2 (15 per 100,000) and 4 (15.4 per 100,000), and during the third wave, this was the case for cluster SE 1 (41.4 per 100,000). In wave 4, the highest percentage of mortality occurred for SE cluster 3 (26.8 per 100,000), while in wave 5, the highest mortality percentage was observed for cluster 2 (15.9 per 100,000). Note that cluster 4 is the smallest SES cluster in terms of the number of confirmed cases per cluster, so in absolute numbers, this relates to a small number of deaths.

Figure 4 depicts the relative risk in each cluster (1–3), where cluster 4 served as a reference group, for each of the variables (confirmed cases, hospitalization, and mortality rate) during the five waves of the pandemic. The Z-test revealed that the disparities between the four SES groups were statistically significant for the number of confirmed cases (*p* < 0.001) and the rates of hospitalization (*p* < 0.001) and mortality (*p* < 0.05) across all five waves.

During the first wave, the highest relative risk of mortality compared to cluster 4 (3.03) was observed in SE cluster 1. In the next wave, the picture turned upside down as the lowest relative risk of mortality compared to cluster 4 (0.73) was observed in SE cluster 1, although the relative risk for the confirmed rate compared to cluster 4 was the highest (29.0). The relative risk for hospitalization compared to cluster 4 was the highest in cluster 1 in the third and fifth waves (6.89 and 5.51, respectively). No similar trend was observed for COVID-19-associated mortality. In wave 3, cluster 1 had the highest mortality rate, while in the cases of wave 4 and wave 5, this was the case for clusters 3 and 2, respectively. The 95% Confidence Intervals (CI) for the Relative Risk of each cluster (1–3) and for cluster 4 (reference) for each wave for confirmed, hospitalized, and death cases are displayed in Table 2.

## 4. Discussion

The findings of this nationwide study cautiously suggest that the socioeconomic gradient patterns previously established in COVID-19 literature [9,10,11,12,13,14,15,16,17,18,19,20] are not applicable to Israel throughout five waves of the SARS-CoV-2 pandemic regarding infections, hospitalizations, and mortality. Furthermore, we show that there has been a gradual decline in the case fatality ratio (CFR) throughout the ongoing outbreak in Israel. This is a strong indicator of the severity of disease and quality of healthcare, as this occurred despite a significant increase in the number of confirmed cases during the fifth wave of the pandemic, which was dominated by the Omicron variant [30]).

The above findings pose a striking curveball to the long-standing and wide-ranging observed health disparities associated with social determinants [10,11,12,13,14,15,16,17,18,19,20,21], though caution should be exercised when interpreting them. Israel seems to present a suitable setting for further investigation of the relationship between SES and health due to the multicultural and multiethnic characteristics of its population, its highly developed national healthcare system, and its universal national health insurance to which all members of the population are entitled. Previous findings have established that health inequality associated with socioeconomic status imposes a significant economic burden on the State of Israel, despite all permanent residents being insured for basic medical services under the National Health Insurance Law [31]. Additionally, disparities have been recognized between minority population groups among Israeli citizens, whereas Jews from the Soviet Union and Arabs were found to have worse health outcomes than nonimmigrant Jews [32]. Both groups have lower documented socioeconomic status compared to the Jewish majority population and differences in patterns of healthcare service utilization have been shown [33]. Structural barriers to healthcare in Israel include costs, transportation difficulties, and language barriers [34]. Furthermore, nonequal distributions of community physicians, hospital beds, and facilities throughout the country have been documented [35]. The findings of this study paint a slightly different picture.

The impacts of SES disparities on health have been more consistently identified in the context of chronic conditions as compared to acute health conditions [36]. This may help to explain our findings and the lack of uniformity in the recurrent theme presented in public health literature that, during a pandemic, low SES groups will be hit harder. Rather, the findings indicate a much more dynamic and complex picture, where not one group dominates the realm of improved outcomes or bears the burden of disease.

In the context of our study, it is known that the first patient with a confirmed COVID-19 infection arrived in Israel from the “Diamond Princess”, a quarantined cruise ship, on 21 February 2020 [37]. Similar to other countries, the first cases of COVID-19 in Israel were traced back to individuals who had returned from overseas travel. This meant that, during the initial stages of the outbreak, the majority of infections were reported in relatively young and well-off individuals [38]. Subsequent infections and deaths in the first wave impacted those from the lowest SES group. Once the existence and dangers of the pandemic had become public knowledge, people and governments adopted precautionary measures through a combination of stay-at-home and social distancing rules, encouraging people to avoid going outside (with the exception of defined essential activities) and to maintain social distancing from individuals outside their own household, alongside mask-wearing efforts [37]. Later waves were more widespread and affected a broader range of individuals. In the second wave, the highest numbers of infections and hospitalizations were observed in the lowest SES cluster, while deaths were more frequently observed in the highest cluster. In the third wave, all three COVID-19 indicators were highest in the lowest SES cluster. As vaccinations became available in the third wave and were rolled out in Israel on 20 December 2020, low SES groups demonstrated significantly lower rates as compared to high SES groups in terms of their vaccination uptake, despite the free availability of vaccines [39]. In this wave, larger gaps in the numbers of infections, hospitalizations, and deaths were observed between SES clusters. In the fourth wave of the pandemic, a booster campaign was rolled out; however, as per our findings, here, the third SES cluster was most afflicted. Furthermore, as the fifth wave is still ongoing, conclusions are still limited and must be made with caution; however, to-date, incongruence between infections, hospitalizations, and deaths has been observed between SES groups.

Additional explanations for the differences observed in hospitalization and death rates across socioeconomic clusters may be explained by variations in the timing of the peak incidence of large-scale epidemic curves [40,41]. During the first four waves, natural immunity was highly protective, resulting in an up-and-down pattern of hospitalizations and deaths for each socioeconomic cluster. As different clusters built up their natural immunity at different rates, the timing and magnitude of peak hospitalizations and deaths varied across clusters. This dynamic effect may account for the inconsistent variability observed across socioeconomic clusters when each wave is examined in isolation. Additionally, lower socioeconomic groups may have experienced an earlier peak in their epidemic curve due to their higher likelihood of being exposed to COVID-19 early on (with the exception of the first wave due to travel abroad), possibly due to being employed in jobs that did not allow for remote work. Figure 2 and Figure 3 of the paper reflect this pattern. Moreover, the natural immunity acquired from the previous waves was compromised by the omicron variant, which is likely to have affected the overall up-and-down pattern to some degree during the fifth wave. This disruption is evident in the incidence of hospitalization cases, with SE cluster 1 being the most affected, as depicted in Figure 2. However, no similar disruption can be seen in the incidence of mortality, as illustrated in Figure 3. This may be due to the less lethal nature of the omicron variant, which was in circulation during the last wave, compared to the variants that circulated during the previous four waves.

Our findings indicate that health trends cannot necessarily be generalized to all countries and are very much dynamic and contingent on the socio-geographical context. In addition, it is necessary to consider the specific characteristics of the disease. While health behavioral differences attributed to different SES groups may partially explain differences in health outcomes (for example, during the third wave where vaccinations were introduced and lower rates of vaccination were observed in lower SES groups), COVID-19 does not discriminate between rich and poor, demonstrating a much more complex picture which must be analyzed in-depth. The implications of this study include the importance of identifying the dynamic interplay and interactions of sociodemographic characteristics and health behavior in order to better improve efforts to enhance health outcomes by policy makers throughout the varied phases of the pandemic. Our results highlight the need for targeted interventions that address the unique health needs and challenges faced by individuals and communities with varying levels of SES. This may include efforts to improve access to healthcare, promote healthy lifestyle behaviors, and address the environmental and social determinants of health.

Furthermore, this study aims to serve as a steppingstone for researchers to explore alternative explanations for the observed dynamic trends alongside the multiple dimensions that may contribute to health outcomes. Further research is necessary to better elucidate the multidimensional factors that contribute to health behavior and outcomes.

Several limitations must be considered when cautiously considering the findings of this study. First, data were analyzed by locality, rather than by individuals, as individual data were not available. Using Socioeconomic Status (SES) at the local level as a proxy measure for an individual’s SES has several important limitations. Firstly, at the local level, SES may not accurately reflect the SES of all individuals living in that area, as some may have a different SES from the general population. Secondly, SES is a complex and multidimensional concept that cannot be captured by a single variable such as income or education level. This means that using an area-level SES measure may not capture the full extent of disparities in health outcomes within a given population. Thirdly, SES may vary within a local area, and using only one proxy measure may lead to the oversimplification and misclassification of individuals. Additionally, the methods used to measure SES may vary across studies, leading to inconsistent results and limitations in comparability. Therefore, it is important to consider the limitations of using SES at the local level as a proxy measure when interpreting the current research findings and making conclusions based on them. Furthermore, in the MOH database, some information on COVID-19 deaths is missing, as the values did not perfectly align with the number of to-date demarcated deaths in the population. In addition, given that data are unavailable for defining symptomatic versus asymptomatic cases, it is possible that asymptomatic testing was performed for social rather than medical reasons and could have varied across the different socioeconomic clusters. Consequently, as the case count includes both symptomatic and asymptomatic cases, we cannot assume that the ratios between symptomatic and asymptomatic cases were identical across the four socioeconomic clusters or across the five waves. Lastly, as the pandemic is still ongoing, the data are not definitive.

## 5. Conclusions

Our findings suggest that health trends cannot be universally applied to all countries and are subject to change depending on the socio-geographical context. It is crucial to consider the unique characteristics of each disease to inform effective public health measures. While health behaviors influenced by socioeconomic status may explain some variations in health outcomes, the pandemic has revealed a more intricate situation that requires a thorough analysis. Our study demonstrates the significance of identifying the complex interplay between sociodemographic characteristics and health behaviors to develop targeted and successful health policies throughout various stages of the pandemic. Moreover, this research is intended to pave the way for future investigations into alternative factors that shape dynamic health trends and their multifaceted effects on health outcomes. Further research is warranted to gain a better understanding of the multidimensional factors contributing to health behaviors and outcomes. In addition, future research could explore the use of more comprehensive measures of SES to capture a broader range of factors that contribute to individual- or population-level vulnerability.

## Figures and Tables

**Figure 1 healthcare-11-00933-f001:**
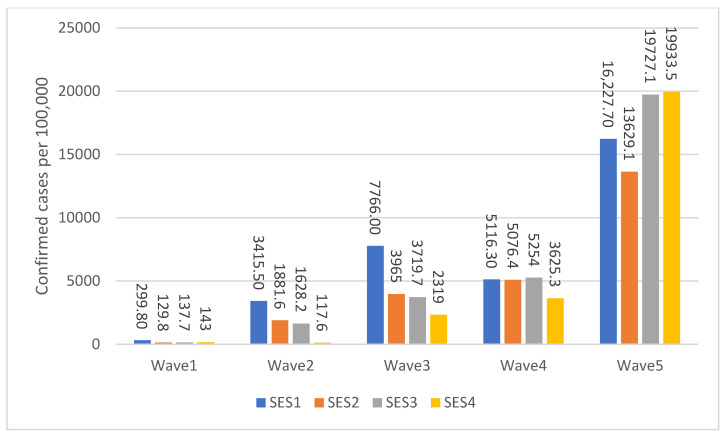
Age-Adjusted Confirmed cases by SES and Waves per 100,000 population.

**Figure 2 healthcare-11-00933-f002:**
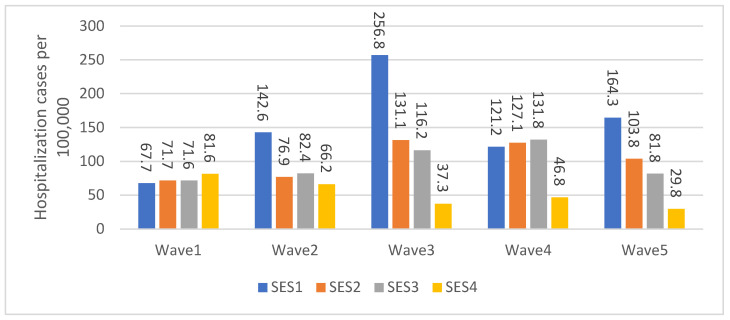
Age-Adjusted hospitalization cases by SES and Waves per 100,000 population.

**Figure 3 healthcare-11-00933-f003:**
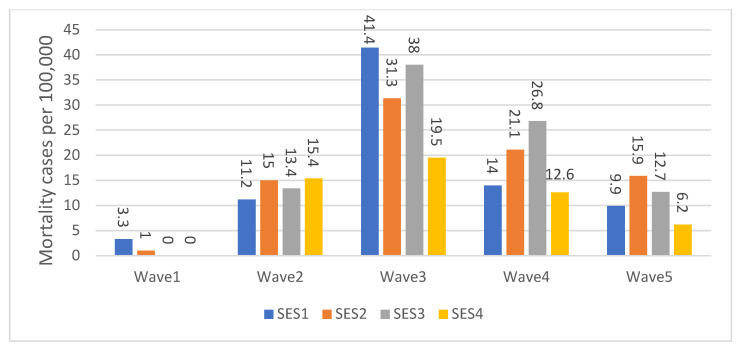
Age-Adjusted death cases by SES and Waves per 100,000 population.

**Figure 4 healthcare-11-00933-f004:**
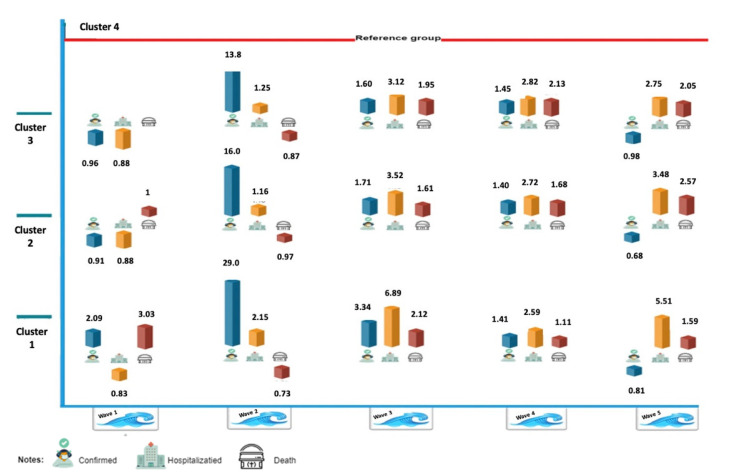
Relative risk between each cluster (1–3) and cluster 4, for each of the variables during the five waves of the pandemic.

**Table 1 healthcare-11-00933-t001:** Overall snapshot of epidemiological data on COVID-19 cases in Israel through five waves of the pandemic.

Wave	1	2	3	4	5
Months	February–May 2020	June–November 2020	December 2020–April 2021	May–November 2021	December 2021–February 2022
**Total number of confirmed cases**	17,124	297,526	523,931	431,642	11,630,353
**Highest number of confirmed cases per day**	740	9078	10,114	11,333	85,141
**Deaths**	289	2281	3813	1618	1110
**Highest number of death cases per day**	13	47	76	36	59
**Case fatality Ratio (Ratio between death/confirmed cases)**	0.02	0.007	0.007	0.003	0.00009
**Severe illness**	643	7989	12,690	5403	3499
**Highest number of severe cases per day**	192	897	1190	767	1254
**Highest severe cases per day (accumulated)**	34	161	193	118	232
**Ratio between severe illness/confirmed cases**	0.03	0.02	0.02	0.01	0.0003
**The highest number of tests**	13,289	67,870	124,663	414,702	474,835
**The highest percentage of positive tests per day**	10.89%	15.52%	10.19%	8.42%	22.96%
**Highest number of active cases per day**	9808	72,400	84,784	92,270	537,755
**Vaccination**	No	No	Middle of Wave	Yes	Yes
**Lockdown**	4.5.2020–25.3.2020	17.10.20–18.9.2020	7.2.2021–27.12.2020	No	No

**Table 2 healthcare-11-00933-t002:** The 95% Confidence Intervals (CI) (left) for the Relative Risk of each cluster (1–3) and cluster 4 (reference) for each wave for confirmed, hospitalized, and death cases and *p*-values (right).

Cluster	Wave	1		2		3		4		5	
**1**	**Confirmed Cases**	1.960–2.233	<0.001	29.003–29.084	<0.001	3.322–3.376	<0.001	1.378–1.444	<0.001	0.796–0.833	<0.001
	**Hospitalized Cases**	0.113–0.687	<0.001	−0.127–0.267	<0.001	1.913–2.207	<0.001	1.494–2.186	<0.001	6.473–7.067	<0.001
	**Death Cases**	−2.195–2.195	<0.001	−1.121–1.171	<0.001	0.0402–1.228	<0.001	0.093–1.481	0.033	1.267–2.655	0.007
**2**	**Confirmed Cases**	0.701–1.115		15.946–16.054		1.672–1.747		1.367–1.433		0.664–0.704	
	**Hospitalized Cases**	0.693–1.247		−0.198–0.338		1.855–2.265		1.603–2.277		4.717–5.463	
	**Death Cases**	−3.802–3.802		−0.921–1.043		0.256–1.622		0.502–1.890		3.057–4.445	
**3**	**Confirmed Cases**	0.762–1.164		13.787–13.903		1.565–1.643		1.417–1.482		0.973–1.006	
	**Hospitalized Cases**	0.633–1.187		−0.170–0.350		1.707–2.413		1.609–2.271		2.350–3.190	
	**Death Cases**			−0.992–1.118		0.598–1.832		0.774–2.162		3.010–4.492	

## Data Availability

The data that support the findings of this study are available from the Ministry of Health Israel COVID-19 Dashboard at https://datadashboard.health.gov.il/COVID-19/general?utm_source=go.gov.il&utm_medium=referral, accessed on 15 January 2023. These data were derived from the following resources available in the public domain: Traffic light by locality.

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
