# Peer review of "Dynamic Trends in Sociodemographic Disparities and COVID-19 Morbidity and Mortality—A Nationwide Study during Two Years of a Pandemic"

_healthcare, 2023, doi:10.3390/healthcare11070933_

Round 1
Reviewer 1 Report
The manuscript titled “Dynamic trends in sociodemographic disparities and COVID-19 morbidity and mortality—a nationwide study during two years of a pandemic” presents a study that evaluated differences in the impact of covid at different socio-economic levels in Israel. The study is important but has some critical issues that need to be resolved before it can be considered for publication.
The presentation is a little confusing.
Add graphical marks to the graphs that indicate statistically significant differences. Specify in the text if the differences found are statistically significant.
On lines 119-121 it is written that “Accordingly, the age-adjusted SES data was utilized for calculation of confirmed cases, hospitalization cases, and death cases. All measures were calculated as a rate per 100,000 individuals in the population.” but the death rate per 100,000 population is not reported in the text: please add it.
The journal standard requires that table captions should appear just after the table. In the current formatting there is confusion between the body text and the caption of the tables.
On lines 159-160 it is written “Figure 2 depicts the percent of hospitalization by four SES categories and 10 clusters, per each month.” but the data in the graph does not refer to the months. Please correct.
No reference is made to Figure 3 in the text.
The data of lines 187 193 must be reported in a table to make them easier to understand.
As reported by the authors, COVID-19 does not discriminate between rich and poor, demonstrating a much more complex picture. On the other hand, there is not only the socio-economic level as a parameter of mortality risk: a study conducted in Italy (DOI 10.3390/healthcare10091684), a country where there is universal health coverage, has shown that health professionals have been exposed to a significant increase in mortality (+ 222% in some cases vs +33% in the general population). Referring to this example, authors should include this point in the introduction.
Author Response
Reviewer #1
Comment#1: The manuscript titled “Dynamic trends in sociodemographic disparities and COVID-19 morbidity and mortality—a nationwide study during two years of a pandemic” presents a study that evaluated differences in the impact of covid at different socio-economic levels in Israel. The study is important but has some critical issues that need to be resolved before it can be considered for publication.
Response#1: Thank you very much for your time and precise editing. We hope this round of revisions meet your concerns.
Location#1: N/A
Comment#2: The presentation is a little confusing.
Response#2: The manuscript has slightly been reworded to ensure better clarity.
Location#2: Throughout the manuscript.
Comment#3: Add graphical marks to the graphs that indicate statistically significant differences. Specify in the text if the differences found are statistically significant.
Response#3: This has now been added to the text, as well as a table (Table 2) to indicate where statistically significant differences have been observed between SES groups, per each wave, for confirmed, hospitalized and death cases. Given the already very busy figures (with text/ color) after attempted visualization with graphical marks as * for statistical significance, it made the Figures more difficult to read, and thus we resort to a table.
Location#3:Results
Comment#4: On lines 119-121 it is written that “Accordingly, the age-adjusted SES data was utilized for calculation of confirmed cases, hospitalization cases, and death cases. All measures were calculated as a rate per 100,000 individuals in the population.” but the death rate per 100,000 population is not reported in the text: please add it.
Response#4: This data is presented in Figure 3, however we have added this to the text as well.
Location#4: Results
Comment#5: The journal standard requires that table captions should appear just after the table. In the current formatting there is confusion between the body text and the caption of the tables.
Response#5:. The formatting has now been adjusted throughout the results section.
Location#5: Results
Comment#6: On lines 159-160 it is written “Figure 2 depicts the percent of hospitalization by four SES categories and 10 clusters, per each month.” but the data in the graph does not refer to the months. Please correct.
Response#6: This has been corrected to “per each wave”.
Location#6: Results
Comment#7: No reference is made to Figure 3 in the text.
Response#7: This has now been updated.
Location#7: Results
Comment#8: The data of lines 187 193 must be reported in a table to make them easier to understand.
Response#8: This data has now been put in a table (Table 2).
Location#8: Table 2.
Comment#9: As reported by the authors, COVID-19 does not discriminate between rich and poor, demonstrating a much more complex picture. On the other hand, there is not only the socio-economic level as a parameter of mortality risk: a study conducted in Italy (DOI 10.3390/healthcare10091684), a country where there is universal health coverage, has shown that health professionals have been exposed to a significant increase in mortality (+ 222% in some cases vs +33% in the general population). Referring to this example, authors should include this point in the introduction.
Response#9: This point has now been added to the introduction as part of the literature review.
Location#9: Introduction
Reviewer 2 Report
The authors study the incidence of COVID-19 cases, hospitalizations, and deaths in Israel during each of five epidemic waves and across distinct community clusters, stratified with respect to social economic status into four socioeconomic levels. The authors have done a good job with literature review as well as discussing the results in the context of previous research literature. The reported results are interesting because they are quite perplexing, as is apparent from the paper's discussion section. Although statistically significant differences are established between the different socio-economic clusters, those differences are not consistent across the 5 COVID-19 epidemic waves, and they appear to defy explanation.
It is possible that what we are looking at for each socioeconomic cluster could be as simple as large-scale epidemic curves that peak differently. If one considers the first four waves, during which it is known that natural immunity remained highly protective [1], one sees that the incidence of hospitalizations and deaths over the first four waves follows an up and down pattern, for each distinct socioeconomic cluster. As different socioeconomic clusters build up natural immunity at variable rates, hospitalizations and deaths peak at different waves and have variable variance. This could be the dynamic effect that explains the inconsistent variability across socioeconomic clusters when each wave is viewed in isolation. Furthermore, if we assume that the lowest socioeconomic clusters tend to have employment that does not allow them to work from home, they would be more likely to be exposed to COVID-19 early on, and thus their corresponding epidemic curve, across all four waves, would peak ahead of the epidemic curves for the higher socioeconomic groups. This is indeed the pattern that I see in Figure 2 and Figure 3 of the paper.
It is known that the omicron variant broke through the natural immunity acquired from the previous waves [2], and that should disrupt the overall up and down pattern to some extent during the 5th wave. This disruption has been observed in the incidence of hospitalization cases shown in figure 2, where SE cluster 1 is again the one most affected. It is not seen in the incidence of mortality shown in figure 3, and that could be explained because of the omicron variants, circulating during the last wave, being less lethal than the variants circulating during the previous four waves.
The above interpretation is, of course, just a hypothesis, but it is an example of a dynamic that could explain the observed results. In countries as large as the United States, the epidemic curves peaked differently by geographic location, and the same is true on a global scale across different nations. For a country as geographically small as Israel, the geographic effect could be negligible enough to allow a similar effect across different social economic clusters to become dominant, which can be explained in terms of variability in the ability of these distinct groups to isolate during the pandemic.
The authors are encouraged to comment about this in the discussion section of their manuscript. It is likely that some readers will consider this as a possible explanation, like I did, and would be interested to know what the authors think about it.
As far as the details of the manuscript are concerned, one point that requires clarification is whether the cases counted are symptomatic only or whether they are also including a count of asymptomatic cases. For the purpose of assessing hospitalization and mortality risk, the denominator should include only symptomatic cases, because what everyone wants to know is what are their chances for hospitalization or death when they are actually symptomatically sick. It is possible that the incidence of asymptomatic testing, performed for social rather than medical reasons, could be variable across the different socioeconomic clusters. Consequently, if the case count includes both symptomatic and asymptomatic cases, then we cannot assume that the ratio between the two is identical across the four socio-economic clusters. Likewise, we cannot assume that this ratio is identical across the 5 distinct epidemic waves either. This is a limitation of the study that affects the results of Figure 1, and the calculation of the Case Fatality Rate given in Table 1, although it shouldn't be a concern with respect to the results of Figure 2 and Figure 3.
The authors should clarify in the manuscript whether the case count includes only symptomatic cases or whether it is both symptomatic and asymptomatic. If data is available, I recommend using symptomatic cases. If data is available to distinguish between symptomatic and asymptomatic cases but there are also cases reported with unknown status, then the best one can do would be to calculate the lower bound and upper bound for the total count of symptomatic cases. This is a question worth exploring depending on the extent of data availability.
The numbers on top of the bars on Figure 3 are easy to read, however I find them more difficult to read in figures 1, 2, and 4, due to the colors used. That should be easy to correct. Likewise, the icons on Figure 4 are difficult to read as well. It may be easier to use larger text, instead of icons, as was done with Figures 1, 2, 3.
The authors should carefully review the references and correct some of them. In particular, references 5, 8, 9, 23, 27, 28, 35, 39 appear to be incomplete. Some of them are probably internet publications that are missing the URL and when they were accessed. Others appear to be journal publications missing some of the essential attributes (journal, volume, pages or article id). Journal names are italicized in some publications but not others.
Also, the manuscript should be checked for use of English.
In light of the above, I recommend a minor revision.
References
1. Murchu, E.; Byrne, P.; Carty, P.; Gascun, C.D.; Keogan, M.; O’Neill, M.; Harrington, P.; Ryan, M. Quantifying the risk of SARS-CoV-2 reinfection over time. Rev. Med. Virol. 2022, 32, e2260.
2. Khan, K.; Karim, F.; Cele, S.; San, J.; Lustig, G.; Tegally, H.; Bernstein, M.; Ganga, Y.; Jule, Z.; Reedoy, K.; et al. Omicron infection enhances neutralizing immunity against the Delta variant. Nature 2022, 607, 356–359
Author Response
Reviewer #2
Comment#1: The authors study the incidence of COVID-19 cases, hospitalizations, and deaths in Israel during each of five epidemic waves and across distinct community clusters, stratified with respect to social economic status into four socioeconomic levels. The authors have done a good job with literature review as well as discussing the results in the context of previous research literature. The reported results are interesting because they are quite perplexing, as is apparent from the paper's discussion section. Although statistically significant differences are established between the different socio-economic clusters, those differences are not consistent across the 5 COVID-19 epidemic waves, and they appear to defy explanation.
Response#1: Thank you very much for your time and precise editing. We hope this round of revisions meet your concerns.
Location#1: N/A
Comment#2: It is possible that what we are looking at for each socioeconomic cluster could be as simple as large-scale epidemic curves that peak differently. If one considers the first four waves, during which it is known that natural immunity remained highly protective [1], one sees that the incidence of hospitalizations and deaths over the first four waves follows an up and down pattern, for each distinct socioeconomic cluster. As different socioeconomic clusters build up natural immunity at variable rates, hospitalizations and deaths peak at different waves and have variable variance. This could be the dynamic effect that explains the inconsistent variability across socioeconomic clusters when each wave is viewed in isolation. Furthermore, if we assume that the lowest socioeconomic clusters tend to have employment that does not allow them to work from home, they would be more likely to be exposed to COVID-19 early on, and thus their corresponding epidemic curve, across all four waves, would peak ahead of the epidemic curves for the higher socioeconomic groups. This is indeed the pattern that I see in Figure 2 and Figure 3 of the paper.
It is known that the omicron variant broke through the natural immunity acquired from the previous waves [2], and that should disrupt the overall up and down pattern to some extent during the 5th wave. This disruption has been observed in the incidence of hospitalization cases shown in figure 2, where SE cluster 1 is again the one most affected. It is not seen in the incidence of mortality shown in figure 3, and that could be explained because of the omicron variants, circulating during the last wave, being less lethal than the variants circulating during the previous four waves.
The above interpretation is, of course, just a hypothesis, but it is an example of a dynamic that could explain the observed results. In countries as large as the United States, the epidemic curves peaked differently by geographic location, and the same is true on a global scale across different nations. For a country as geographically small as Israel, the geographic effect could be negligible enough to allow a similar effect across different social economic clusters to become dominant, which can be explained in terms of variability in the ability of these distinct groups to isolate during the pandemic.
The authors are encouraged to comment about this in the discussion section of their manuscript. It is likely that some readers will consider this as a possible explanation, like I did, and would be interested to know what the authors think about it.
Response#2: This is a very good potential explanation. We have added this contextualization to the discussion. Thank you very much.
Location#2: Discussion
Comment#3: As far as the details of the manuscript are concerned, one point that requires clarification is whether the cases counted are symptomatic only or whether they are also including a count of asymptomatic cases. For the purpose of assessing hospitalization and mortality risk, the denominator should include only symptomatic cases, because what everyone wants to know is what are their chances for hospitalization or death when they are actually symptomatically sick. It is possible that the incidence of asymptomatic testing, performed for social rather than medical reasons, could be variable across the different socioeconomic clusters. Consequently, if the case count includes both symptomatic and asymptomatic cases, then we cannot assume that the ratio between the two is identical across the four socio-economic clusters. Likewise, we cannot assume that this ratio is identical across the 5 distinct epidemic waves either. This is a limitation of the study that affects the results of Figure 1, and the calculation of the Case Fatality Rate given in Table 1, although it shouldn't be a concern with respect to the results of Figure 2 and Figure 3.
Response#3: Due to unavailability in the dataset from the Ministry of Health, it is not possible to distinguish between symptomatic and asymptomatic cases (this is now clarified in the methods section). These factors have now been listed as limitations.
Location#3: Methods / Limitations
Comment#4: The authors should clarify in the manuscript whether the case count includes only symptomatic cases or whether it is both symptomatic and asymptomatic. If data is available, I recommend using symptomatic cases. If data is available to distinguish between symptomatic and asymptomatic cases but there are also cases reported with unknown status, then the best one can do would be to calculate the lower bound and upper bound for the total count of symptomatic cases. This is a question worth exploring depending on the extent of data availability.
Response#4: Given that this data is unavailable in the MOH database, we have identified in the methods section that the data includes both symptomatic and asymptomatic cases, and thus limitations for interpretability exist (now listed in the limitations section).
Location#4: Methods/ Limitations
Comment#5: The numbers on top of the bars on Figure 3 are easy to read, however I find them more difficult to read in figures 1, 2, and 4, due to the colors used. That should be easy to correct. Likewise, the icons on Figure 4 are difficult to read as well. It may be easier to use larger text, instead of icons, as was done with Figures 1, 2, 3.
Response#5: The figures have now been adjusted to ensure for better readability and clarity.
Location#5: Results
Comment#6:T he authors should carefully review the references and correct some of them. In particular, references 5, 8, 9, 23, 27, 28, 35, 39 appear to be incomplete. Some of them are probably internet publications that are missing the URL and when they were accessed. Others appear to be journal publications missing some of the essential attributes (journal, volume, pages or article id). Journal names are italicized in some publications but not others.
Response#6: This has now been adjusted for the references.
Location#6: References
Reviewer 3 Report
Thank you for providing an opportunity to review the manuscript. This is an interesting study that provides the impact of socioeconomic status on morbidity, hospitalization and mortality outcomes throughout five waves of the pandemic amongst the Israeli population. The text is relatively well written; however, it needs major improvement:
1) In Abstract, author should add the specific benefit point of this study to the public policy or how to apply for decision maker.
2) In the introduction, please update the covid-19 situation. Please give more example of socioeconomic related with each morbidity, hospitalization and mortality outcomes
3) In method, please add how to analyse the difference of proportion of confirmed cases, hospitalized, death in each SES within and between waves using Z test for discrete data.
4) In results, please add the results of analysing the difference of proportion of confirmed cases, hospitalized, death in each SES within and between waves using Z test for discrete data.
5) Major concerns are typically, ANOVA is used for continuous data, but discrete data are also common in practice (However, it is incorrect to use the F test for continuous data for ANOVA with binary or count data (discrete data)).
6) Please explain each colour in each plot under the figure.
7) Figure 4 should be changed to table and show both relative risk value and confidence interval to make it clearer and easier to the reader.
8) In the discussion, please include and compare with previous studies in the other countries and continents.
9) In conclusion, authors should present and explain more on the benefit point of this study to the public policy or how to apply for decision maker.
Author Response
Comment#1: Thank you for providing an opportunity to review the manuscript. This is an interesting study that provides the impact of socioeconomic status on morbidity, hospitalization and mortality outcomes throughout five waves of the pandemic amongst the Israeli population. The text is relatively well written; however, it needs major improvement:
Response#1: Thank you very much for your time and precise editing. We hope this round of revisions meet your concerns.
Location#1: Throughout the manuscript
Comment#2 In Abstract, author should add the specific benefit point of this study to the public policy or how to apply for decision maker.
Response#2: Per your suggestion, this has now been adjusted in the abstract.
Location#2: Abstract
Comment#3: In the introduction, please update the covid-19 situation. Please give more example of socioeconomic related with each morbidity, hospitalization and mortality outcomes
Response#3: We have provided additional in the introduction section that have been tied to socioeconomic factors to increase morbidity, and more severe outcomes. In addition we have updated the current covid-19 situation in the introduction.
Location#3: Introduction
Comment#4: In method, please add how to analyse the difference of proportion of confirmed cases, hospitalized, death in each SES within and between waves using Z test for discrete data.
Response#4: Thank you for this important comment. We added the following sentence to the methods section: “We analyzed the proportional difference of confirmed, hospitalized, and death cases in each SES within and between waves using Z test for discrete data. “
Location#4: Methods.
Comment#5: In results, please add the results of analysing the difference of proportion of confirmed cases, hospitalized, death in each SES within and between waves using Z test for discrete data.
Response#5: Thank you for your valuable feedback. We have taken your suggestion into consideration and have conducted further analysis to include the results of analyzing the difference of proportions of confirmed cases, hospitalized, and death in each SES using Z test for discrete data.
Location#5: Results
Comment#6: Major concerns are typically, ANOVA is used for continuous data, but discrete data are also common in practice (However, it is incorrect to use the F test for continuous data for ANOVA with binary or count data (discrete data)).
Response#6: Thank you for pointing out that ANOVA may not be suitable for discrete data, and that the F test should not be used for continuous data in ANOVA with binary or count data
Based on your suggestion and further statistical consultation, we have analyzed the difference of proportion using Z test for discrete data.
Location#6: Methods/ Results
Comment#7: Please explain each colour in each plot under the figure.
Response#7: The legends have now been added to ensure clarity.
Location#7: Figure 1-3.
Comment#8: Figure 4 should be changed to table and show both relative risk value and confidence interval to make it clearer and easier to the reader.
Response#8: Th figure has partially been adapted to a table in order to ensure easier readability/ clarity. The figure resolution has also been updated.
Location#8: Results
Comment#9: In the discussion, please include and compare with previous studies in the other countries and continents.
Response#9:. We have added reference to previous studies that indicate differing results to ours in the discussion. In addition, we add an additional potential explanation worthy of considering with respect to the current findings and supporting literature.
Location#9: Discussion
Comment#10: In conclusion, authors should present and explain more on the benefit point of this study to the public policy or how to apply for decision maker.
Response#10: The benefit of such a study is mostly directed to researchers, however the long-term implications of this study and additional epidemiological research being conducted is in deciphering trends and observations in order to assist in the development of tailor-made interventions (whether it be in implementing such interventions among specific groups) or in the context of the current findings, directing interventions to the varied SES groups at varied relevant time-points to ensure improved health outcomes. This has now been expanded upon in the discussion.
Location#10: Discussion
Round 2
Reviewer 1 Report
The name of the ordinate axis is wrong in graphs 2 and 3.
For the rest, the authors have replied satisfactorily
Author Response
Comment#1: The name of the ordinate axis is wrong in graphs 2 and 3.
Response#1: This has now been edited.
Location#1: Graphs 2 and 3
Comment#2: For the rest, the authors have replied satisfactorily
Response#2: Thank you very much for your precise editing and time.
Location#2: N/A
Reviewer 3 Report
Thank you for providing an opportunity to review the manuscript. This is an interesting study that provides the impact of socioeconomic status on morbidity, hospitalization and mortality outcomes throughout five waves of the pandemic amongst the Israeli population. Thank you to the authors for the tremendous amount of work they have done to improve the manuscript. However, it needs minor improvement:
1) In the result, table 2 should be adjusted the value of 95% Confidence Intervals (CI) (right) per each wave by changing it into the 95% CI of Relative risk of each cluster for confirmed, hospitalized and death cases.
2) In the result, an author should explain more about RR of each wave.
Author Response
Comment#1 Thank you for providing an opportunity to review the manuscript. This is an interesting study that provides the impact of socioeconomic status on morbidity, hospitalization and mortality outcomes throughout five waves of the pandemic amongst the Israeli population. Thank you to the authors for the tremendous amount of work they have done to improve the manuscript. However, it needs minor improvement:
Response#1: Thank you very much for your time and precise editing. We hope this next round of edits meet your concerns.
Location#1: Throughout the manuscript
Comment#2: In the result, table 2 should be adjusted the value of 95% Confidence Intervals (CI) (right) per each wave by changing it into the 95% CI of Relative risk of each cluster for confirmed, hospitalized and death cases.
Response#2: This has now been redacted.
Location#2: Results
Comment#3: In the result, an author should explain more about RR of each wave.
Response#3: Relative risk for each wave is now further expanded upon.
Location#3: Results